# How Do Classical Subtypes Correspond to Endotypes in Atopic Dermatitis?

**DOI:** 10.3390/ijms25010265

**Published:** 2023-12-23

**Authors:** Tsuyoshi Suzuki, Shumpei Kondo, Yasuaki Ogura, Masaki Otsuka, Yoshiki Tokura

**Affiliations:** 1Department of Dermatology & Skin Oncology, Chutoen General Medical Center, 1-1 Shobugaike, Kakegawa 436-8555, Japan; ts180chiga@gmail.com (T.S.); knd.shnp@gmail.com (S.K.); y.ogura523@gmail.com (Y.O.); masaki-o@chutoen-hp.shizuoka.jp (M.O.); 2Allergic Disease Research Center, Chutoen General Medical Center, 1-1 Shobugaike, Kakegawa 436-8555, Japan

**Keywords:** atopic dermatitis, endotype, extrinsic type, intrinsic type, subtype, phenotype

## Abstract

Since atopic dermatitis (AD) is a heterogeneous condition, the subtyping of AD is a crucial issue. The classical subtypes of AD are represented by extrinsic and intrinsic subtypes, European–American and Asian subtypes, and adult and pediatric subtypes. While the subtyping of AD was historically conducted based on the phenotype, recent findings on the mechanisms of AD have revealed the importance of the endotype, which can characterize individual patients more accurately. Considering the current development of AD therapies, AD endotyping is a prerequisite for a personalized therapeutic choice. Endotypes of AD can be stratified from different viewpoints, including cytokine expression patterns, allergen properties, epidermal barrier conditions, ceramide variation, the involvement of innate immunity, and serum biomarkers. Among them, the cytokine-based endotype seems to be the most useful one and is categorized into type 2 cytokine (IL-4, IL-13 and IL-31)-high, type 1 cytokine (IFN-*γ*)-high, and/or type 3 cytokine (IL-22 and IL-17)-high, or mixed subtypes. Recently proposed biomarker endotyping aims at individualized treatment options, although the daily clinical use of endotypes is a future issue. To better understand the endotypes for clinicians, attempts to adjust each of the classical subtypes to endotypes are required. This review will discuss the correspondence of the classical subtypes to the various endotypes that have recently been proposed.

## 1. Introduction

One of the current issues in atopic dermatitis (AD) is how to divide this complicated disease into subtypes [1,2,3,4]. Since AD is a heterogeneous condition, it is a requirement to comprehend the pathogenesis of AD by subtyping. Moreover, to optimize the management of allergic diseases, a personalized therapeutic approach is essential [5]. The categorization of AD may lead to a precision medical approach.

Although phenotypes have been historically used for the subtyping of AD [1], the current concept of endotypes may be of greater clinical importance [1,2,3,4]. Therefore, physicians are willing to establish a personalized therapeutic approach to AD that is ideally based on the endotype. Since a number of new effective drugs have been marketed [6,7] and clinical trials of novel therapies are ongoing, information on endotypic properties in individual patients is required for the choice of medicine [2]. However, endotypes may not necessarily be familiar to clinicians because of the difficult accessibility of them in clinical settings. Physicians usually see AD patients consciously or even unconsciously using classical subtypes. To better understand endotypes, therefore, attempts to adjust individual endotypes to classical subtypes are necessary. In this line of study, Tokura Y presented our viewpoint in the International Eczema Council at the International Society for Investigative Dermatology Meeting held on 10 May 2023 in Tokyo, with open discussion.

Concerning databases, selection methodologies, articles, and review processes, it is not easy to conduct our survey, because the terms “classical subtype”, “phenotype”, and “endotype” are not homogeneously used in individual articles. Since our purpose is to investigate the relationship between endotypes and classical subtypes, we primarily concentrated on endotypes. By using PubMed, we searched for “atopic dermatitis” plus “endotype”. Then, we selected “endotype” AD articles that were related to classical subtypes. In another line of search, “atopic dermatitis”, “subtypes”, and “cytokine”, “barrier”, “innate immunity” or “ceramide” were investigated. These keywords were selected because we highlighted them as the viewpoints of endotypes. It was difficult to further collect intended articles by using the vague word “classical subtype” and the limitless word “phenotype”.

Because of the great heterogeneity of the studies and the different classifications and viewpoints used for describing AD, it is difficult to systematically define the reported data. Therefore, our better solution is to clearly express that the presented data are based on a personal, expert, viewpoint.

## 2. Phenotype, Endotype, and Classical Subtype of AD

One of the fundamental issues in AD subtyping is the definition of subtype nomenclature. Because of the difficulty in terminology, we first orient the “phenotype”, “endotype”, and “classical subtype”.

The phenotype is historically the first methodology for AD subtyping and is represented in the skin. It is the most understandable measure and is easily recognized in daily outpatient care without any specific examination [1].

On the other hand, the endotype represents pathogenetic mechanisms and is somewhat confusing because various viewpoints have been proposed for endotyping AD [1,4,5]. First, skin barrier condition contributes to endotypes. The extent of barrier impairment, in particular, the presence or absence of filaggrin (FLG) gene mutation [8,9] and ceramide variation [10], is a representative element. Another critical viewpoint for endotyping is the immune condition. AD usually shows a type 2-shifted immunological condition, but some patients show an additional involvement of type 1 and/or type 3 (Th17/Tc17 and Th22/Tc22) immune cells [1,2,3,4]. In relation to a skewing immune condition, the type of antigen to which patients are exposed is involved in endotyping. Protein allergens and metal/hapten antigens induce the expression of interleukin (IL)-4/IL-13 and interferon (IFN)-*γ*/IL-22/IL-17, respectively [1,2,3,4]. Moreover, recent attempts at biomarker-based endotyping have also been reported, resulting in a further disoriented use of endotypes. Thus, although the endotype is represented by pathogenetic mechanisms, the broad aspects of viewpoints or elements rather make the endotype ambiguous. To solve this issue, more specific names, such as cytokine endotype and barrier endotype, might be necessary.

In advance of or in parallel with phenotyping and endotyping, classical subtyping has been proposed, including extrinsic and intrinsic subtypes, European–American and Asian subtypes, and pediatric and adult subtypes. These are comprehensive subtypes, and each of the classical subtypes is outlined by phenotypes and endotypes. In other words, classical subtypes are an umbrella, and are unconsciously determined via phenotypes and endotypes.

As other methods to subdivide AD, genotypes, regiotypes, and theratypes have been suggested [5]. However, these types are not independent and may overlap with the other types.

## 3. Phenotypes of AD: Scarce Utilization for Subtyping

Main clinical manifestations of AD include infantile eczema, goose flesh-like skin, eczema on creases, lick dermatitis, dirty neck, Hertoghe sign, Dennie–Morgan folds, red face, and dirty neck [1]. It is difficult for even dermatologists to use these skin lesions for categorization into subtypes because most of them are the phenotype of common extrinsic AD [1,11]. Among them, only Dennie–Morgan fold is exceptional, as its frequency was reported to be higher in intrinsic AD than in extrinsic AD [12]. Prurigo nodularis may be another category of AD phenotypes [13]. When we studied the incidence of prurigo nodularis in Japanese patients with extrinsic and intrinsic AD, both subtypes showed prurigo nodularis at comparable frequencies (data not published), suggesting no preponderance of its occurrence between the classical subtypes.

Notably, palmar and plantar hyperlinearity and ichthyosis vulgaris are frequently seen in extrinsic AD patients and are closely associated with *FLG* mutations and subsequent barrier abnormality [1,11]. Thus, these phenotypic signs are not only of endotypes, but also of genotypes. As less frequent phenotypic skin lesions, lichen amyloidosus, vitiligo or leukoderma, chondrodermatitis of the auricle, and angiohistiocytoid papules have been reported. Since these lesions arise in patients with high serum IgE levels and eosinophilia, they are estimated to be the manifestations of type 2-predominant extrinsic AD. Again, we cannot use even these intriguing lesions for the elaborate subtyping of AD.

When we refer to the historical diagnostic standard of AD by Hanifin and Rajka [14], we can find that elements of both phenotype and endotype are blended in the list. The phenotypic signs include xerosis, ichthyosis, palmar hyperlinearity, hand or foot dermatitis, nipple eczema, cheilitis, recurrent conjunctivitis, Dennie–Morgan folds, keratoconus, anterior subcapsular cataracts, orbital darkening, facial pallor/facial erythema, and pityriasis alba. Meanwhile, the endotypic elements include immediate (type 1) skin test reaction, elevated serum IgE, impaired cell mediated immunity, food intolerance, and clinical course influenced by environmental and emotional factors. Thus, phenotypes and endotypes were historically intermingled and some of them are difficult to be distinguished from each other.

## 4. Classical Subtypes of AD: Relevance to Cytokine Skewing

In the era when the idea of endotypes was not yet apparent, AD was categorized into extrinsic and intrinsic types mainly by serum levels of IgE [11], into European–American and Asian subtypes by ethnicity [15,16], or into pediatric and adult subtypes by age bracket [1,3] (Table 1). It is notable that different immune conditions operate between the counterparts in these three representative classical subtypes (Table 1). In “extrinsic”, “European American”, and “pediatric” subtypes, Th2 (type 2) inflammation is dominant. On the other hand, “intrinsic”, “Asian”, and “adult” subtypes are characterized by various extents of a Th1 (type 1)-, Th22/Th17 (type 3)-, and/or IL-19A (related to type 3 mainly in the presence of autoimmune mechanisms, but also to type 2 in the Th2-dependent allergic process)-skewing condition [1,2,3,4,15,16,17]. A current issue is how these classical subtypes correspond to endotypes. Alternatively, some investigators might feel that endotypes are not necessarily relevant to classical subtypes. In any case, attempts to adjust each classical subtype to an endotype are a prerequisite to better understand endotypes.

### 4.1. Extrinsic and Intrinsic AD

We first describe the serum IgE level-high extrinsic and IgE-normal intrinsic types, which were historically called mixed and pure AD, allergic and non-allergic AD, or classical AD and atopiform dermatitis, respectively [11]. Extrinsic and intrinsic AD began to be adapted in the late 1980s. While extrinsic AD is the common type with a high prevalence, the incidence of intrinsic AD is approximately 20% or less, with a female predominance [11]. The similar frequency rates of the two subtypes have been reported in European and Asian countries, including Germany, Hungary, Netherland, Korea, and Japan [11]. The skin barrier is perturbed in extrinsic AD [18], and FLG mutation represents a typical cause of barrier impairment [19]. Thus, allergic conditions may be preceded by skin barrier abnormalities, which allow protein antigens to penetrate through the disrupted stratum corneum barrier in extrinsic AD. Protein allergens generally induce and evoke Th2 cells, which produce IL-4 and IL-5, thereby elevating IgE and eosinophils.

On the other hand, intrinsic AD is immunologically characterized by the higher expression of IFN-*γ* and IL-22/IL-17 [19,20], although type 2 cytokines, such as IL-4, are also overexpressed because of the background of AD. Non-protein antigens, such as metals (nickel, cobalt, and chromium) and haptens, may induce Th1/Th17/Th22 (type 1 and type 3) responses and evoke eczematous dermatitis [21]. Higher levels of nickel concentration were found in peripheral blood and sweat in intrinsic AD than those in extrinsic AD [21,22]. Some intrinsic AD patients intake high-nickel-containing foods, such as coffee and chocolate [22]. Since protein allergens are not the primary antigen in intrinsic AD, serum total IgE or IgE specific for mite antigens is not elevated. We use tentative criteria as follows: intrinsic AD is defined as serum IgE levels ≤ 200 kU/L or 200 < IgE ≤ 400 plus class 0 or 1 of IgE specific to *Dermatophagoides pteronyssinus* or *Dermatophagoides farinae*, and extrinsic AD is defined as 400 < IgE levels or 200 < IgE ≤ 400 plus class 2 or more of the specific IgE [21].

Although the exact mechanisms underlying intrinsic AD remain unclear, we focused on suprabasin deficiency as one of the causes. The expression of suprabasin was decreased in the stratum corneum of AD patients compared with that in healthy subjects, as determined via proteome analysis [23]. The serum suprabasin in intrinsic AD was significantly lower than that in the control and tended to be lower than that in extrinsic AD [24]. In three-dimensionally constructed skin, suprabasin deficiency induced apoptosis of epidermal keratinocytes [24]. Suprabasin is expressed in the upper digestive tract where nickel is absorbed. In suprabasin-knock-out mice, the impaired expression of suprabasin increased the absorption of orally administered nickel and elevated serum nickel levels. In those nickel-loaded, suprabasin-deficient mice, contact hypersensitivity to nickel was enhanced [25]. These findings suggest that suprabasin deficiency is one of the causes of intrinsic AD. However, there may be other mechanisms underlying intrinsic AD.

### 4.2. European–American and Asian AD

Different T-cell activation profiles have been found between Western and Asian populations [15,16]. In addition to type 2, there is the co-existence of multiple cytokine axes of type 1 and type 3 in Asian types of AD [3]. Accordingly, while Th17 frequencies were elevated in both the peripheral blood and lesional skin of Japanese patients [15], Th17 axis activation was not seen in European–American extrinsic patients [14]. Thus, Asian AD patients are characterized by unique blended immune dysregulation between AD and psoriasis [2,3,15]. Nevertheless, increased levels of Th2 cytokines and related chemokines were similarly seen in European–American and Asian patients, correlating with AD severity and IgE levels [15,16]. A study comparing lesional and nonlesional skin of European–American and Asian (Japanese and Koreans) patients showed prominent epidermal hyperplasia and marked parakeratosis in Asian subjects but relatively preserved barrier proteins, FLG, and loricrin [15,16]. In accordance with these histological findings, the Asian type of AD occasionally shows skin lesions resembling psoriasis (Figure 1). IL-19 is induced by IL-4, IL-13, and IL-17, and it augments IL-17’s effects on keratinocytes. Levels of IL-19 were significantly greater in AD lesions of Asian patients [15,16]. Chinese AD patients share the consistent Th17/Th2 or blended AD–psoriasis type with Japanese and Korean patients [26].

In another ethnic study, African American and European–American patients were compared. Loss-of-function mutations of FLG are not prevalent in African American patients with AD, but FLG variations were associated with AD persistence in these patients [27]. Immunologically, Th1/Th17 attenuation and Th2/Th22 skewing were seen in these patients [28].

### 4.3. Pediatric and Adult AD

In infancy, seborrheic dermatitis is a common eruption on the face and scalp. In young children, dry skin is overt, as represented by goose flesh-like skin, and eczema occurs on the antecubital and popliteal fossae. In adults, lichenified lesions are prominent, and patients may develop red face, exhibiting persistent dark reddish erythema on the face, and dirty neck, showing poikilodermatous reticulate lesions on the neck. These skin lesions may be associated with infiltrating T-cell populations, epidermal changes, and dermal remodeling [1].

In peripheral blood lymphocytes, the frequencies of cutaneous lymphocyte-associated antigen (CLA)^+^ Th1 cells were significantly lower in AD infants than those in older patients, but the frequencies of CLA^+^ Th2 cells were similarly expanded across all AD age groups [29]. After infancy, CLA^−^ Th2 frequencies were increased in all age groups, suggesting systemic immune activation with disease chronicity. IL-22 frequencies serially increased from normal levels in infants to highly significant levels in adolescents and adults [29]. Thus, in adults, the type 3 and type 1 pathways are involved, and a weakened epidermal barrier is characteristic. In contrast, pediatric patients exhibit less type 1 activation, and defects in epidermal lipid metabolism contribute to their barrier defect [4].

In addition to the pediatric and adult subtypes, the early-onset and late-onset subtypes have more recently proposed. Their definition remains unclear, and their usage is different among studies. They are used similarly to the child onset and adult onset, but in some studies, they mean the early-onset and late-onset subgroups in pediatric AD. In two birth cohort studies including 9894 children from the UK and 3652 children from the Netherlands, the most prevalent class was early-onset–early-resolving AD, which was associated with male sex. Two classes of persistent disease were identified (early-onset-persistent and early-onset-late-resolving); these were most strongly associated with the AD genetic risk score as well as the personal and parental history of atopic disease. An as-yet unrecognized class of mid-onset-resolving AD, not associated with *FLG* mutations, but strongly associated with asthma, was identified [30]. *FLG* mutations are associated with only the early onset of childhood AD, not the late onset of it [31]. A Korean group of investigators classified the patients into two groups: early-onset (first onset of AD before 12 years of age) and late-onset (first onset of AD at 12 years of age or later). An association with allergic asthma or rhinitis, a family history of atopic disease, elevated total serum IgE, and sensitivity to food allergens were more commonly seen in the early-onset group. The late-onset group had a significant likelihood of nonflexural involvement. There was no significant difference in the mean eczema area severity index score, eosinophil count, and sensitivity to aeroallergens between the two groups [32].

## 5. Endotypes of AD: Viewpoints for Subtyping

Endotypes of AD can be stratified from different factors, including cytokine patterns, allergen properties, epidermal barrier condition, ceramide variation, the involvement of innate immunity, and serum biomarkers [1,2,3,4]. These factors are mostly viewpoints, but methodologies are intermingled (Table 2). Thus, endotypes are not as simply understandable as phenotypes are. Recent technologies further provide new methodologies for endotyping AD [33,34].

It should be noted that these viewpoints are not independent, but rather confound each other. For example, the expression of individual cytokines is deeply influenced by types of allergens [1,2,3,4,21]. The penetration of allergens through the epidermis depends on epidermal barrier condition, including *stratum corneum* barrier constituents and intracellular lipids [8,10,17]. Innate immunity is closely associated with other factors [31] and overlapped with cytokines. Finally, new methodologies may propose novel endotypes and affect the importance of individual viewpoints [33,34].

### 5.1. Cytokines of Acquired Immunity

Cytokines are one of the most frequently used viewpoints for endotyping AD, and their expression levels can be measured at the protein or mRNA levels in blood and skin specimens from the patients [1,2,3,4,19,20]. Cytokine-based endotypes can be categorized into type 2 cytokine (IL-4, IL-13, IL-5 and IL-31)-high, type 1 cytokine (IFN-*γ*)-high, and/or type 3 cytokine (IL-22 and IL-17)-high, or mixed subtypes. The classical subtypes are largely related to these cytokine-based endotypes (Figure 1).

Along with type 2, the co-existence of multiple cytokine axes of type 3 and type 1 types is seen especially in intrinsic and Asian types of AD. Intrinsic AD shows similar Th2 and higher Th1 and Th17 immune activation compared to that in extrinsic AD [19,20]. The Asian AD phenotype combines features of AD and psoriasis with increased Th17 polarization [15,16,17]. The frequencies of CLA^+^ Th1 were significantly lower in AD infants than those in older patients, but CLA^+^ Th2 was similarly expanded across all age groups [29]. Thus, a type 2 dominant inflammatory state, generally known to be an essential characteristic of AD, is seen in “extrinsic”, “European American”, and “pediatric” subtypes. Taken together, type 3 and type 1 cytokines are additionally accompanied with the type 2 axis in “Asian”, “intrinsic”, and “adult” types (Figure 1).

### 5.2. Allergens and Epidermal Barrier

Endotyping based on allergens is closely associated with epidermal barrier-based endotyping. In epidermal barrier-disrupted skin, protein antigens or allergens can penetrate through the skin and induce type 2 inflammation with Th2 cells and innate lymphoid cell type 2 (ILC2) [35,36]. The barrier damage stimulates epidermal keratinocytes to produce alarmin IL-33, IL-25, and thymic stromal lymphopoietin (TSLP), thereby activating a type 2 reaction and promoting IL-4 production [36]. This sequential event provides the mechanism of extrinsic AD and European–American AD. On the other hand, a relatively preserved epidermal barrier may allow metals and haptens to penetrate the epidermis, as typically seen in intrinsic AD [21].

Ceramides are involved in skin barrier function [10]. It is assumed that ceramides participate in AD endotypes, but the exact contribution of ceramide variation to endotyping will be investigated in future.

### 5.3. Innate Immunity

There is no definite study showing that the involvement of innate immunity differs between the classical subtypes. However, the pathogenetic role of innate immunity for AD is currently at the center of attention [37,38,39,40,41]. Since the requirement of specific antigens remains a controversial matter for AD pathology, innate immunity might differentially contribute to individual AD patients and possibly determines the subtypes.

IL-33 is an inflammatory cytokine that is over-expressed in epidermal keratinocytes of patients with AD [37]. IL-33 transgenic mice, which express IL-33 specifically in keratinocytes, spontaneously develop AD-like eczema, suggesting that IL-33 is sufficient for the development of AD [38]. IL-33 stimulates various cells, including ILC2, to produce type 2 cytokines, such as IL-5 and IL-13, and IL-33-stimulated basophils activate ILC2 via IL-4 [39]. Moreover, the mechanism of IL-4 production by ILC2 has recently been investigated and was elaborated with other cytokines [40]. This raises a possibility that IL-4 production by ILC2 varies among patients, leading to endotyping.

The roles of IL-1 family members, IL-36 and IL-38, are postulated in the pathogenesis of AD [41]. There have been scarce studies on how the production of these cytokines varies in AD individuals. Future studies are necessary to elucidate this issue.

### 5.4. Serum Biomarkers

We mention a representative methodology for endotyping. Biomarkers are valuable parameters for precision medicine as they provide information on disease endotypes, clusters, precision diagnoses, the identification of therapeutic targets, and the monitoring of treatment efficacies.

Thijs JL et al. conducted two cohort studies to investigate AD endotypes by using serum biomarkers [33,34]. The authors used the biomarkers in the peripheral blood. Given that the substances in the skin can flow into the circulation, biomarkers may reflect both the conditions of the peripheral blood and skin. In this sense, the endotypes result from the orchestrated organs including the skin and are meaningful. In both studies, they found four clusters in the endotype, and three of them were virtually identical. In the latest analysis based on 143 serum biomarkers from 146 patients with severe AD, cluster A (33.6%) could be distinguished from the other clusters as being a “skin-homing chemokines/IL-1R1-dominant” cluster, whereas cluster B (18.5%) was a “Th1/Th2/Th17-dominant” cluster, cluster C (18.5%) was a “Th2/Th22/PARC-dominant” cluster, and cluster D (29.5%) was a “Th2/eosinophil-inferior” cluster [34]. Their findings indicate that, in addition to Th2, Th1 and Th17/Th22 are involved in some subgroups of AD, and that even a Th2/eosinophil-inferior type exists. We speculate that cluster A corresponds to a group with high levels of severity-associated chemokines, cluster B corresponds to heavily inflammatory AD, cluster C corresponds to AD with chronic lesions or the Asian type, and cluster D corresponds to intrinsic AD. However, endotyping using serum biomarkers may be fundamentally different from conventional subtyping, and it may be difficult for the endotypes to comprehensively correspond to the classical subtypes.

In a study, the top 10 biomarkers include IL-37, IL-1ra, XCL-1, eotaxin/CCL11, IL-1ß, IL-26, LIGHT/TNFSF14, IL-1r1, EGF and TSLP [34]. The authors were surprised at the fact that none of these markers are Th2-related cytokines, but they consisted of IL-1-, IL-10-, and epithelium-related markers. Their study might indicate that individualized treatment options should be based not on clinical phenotypes of AD but instead on biomarker-based endotypes.

## 6. Possible Therapeutic Application of Endotypes

Endotypes provide important information on individualized treatment options. With an AD patient belonging to type 2-dominant endotypes, anti-IL-4Ra antibody dupilumab, blocking both IL-4R type 1 (for IL-4) and type 2 (for IL-4 and IL-13) [42,43] and neutralizing antibodies against IL-13 activity, tralokinumab [44], and lebrikizumab [45] are effective. These treatments also theoretically exert a beneficial effect on extrinsic AD and European–American AD. Meanwhile, they might be less sufficient efficacious for a Th2/eosinophil-inferior endotype or intrinsic AD. However, clinical trials (Chronos) showed that the therapeutic effect of dupilumab on AD did not depend on the serum levels of IgE or TARC (data from clinical trials; Dupiumab Common Technical Document Part 2—Clinical Overview 2.7.3 Clinical Effectiveness). It would be interesting if this finding could be confirmed in the real world. Furthermore, anti-IL-31Ra antibody nemolizumab is another modality and decreases the pruritus of AD [46].

In addition to these biologics, Janus kinase (JAK) inhibitors, baricitinib [47], upadacitinib [48], and abrocitinib [49] have recently been marketed for the treatment of AD. They mainly inhibit JAK1, which is crucial for the induction of AD [6,7,50] (Figure 2). In fact, JAK1 gain-of-function causes AD-like skin lesions, eosinophilia, and keratinization of the epidermis [51,52,53]. Thus, it seems that JAK1 inhibitors have a therapeutic effect on type 2 inflammation, as represented by extrinsic AD, European–American AD, and pediatric AD. However, JAK1 signaling transduces not only for type 2 cytokines, but also for IFN-α/β, IFN-*γ*, and IL-22 [50,54,55] (Figure 3). Given this broad ability to inhibit type 2, type 1, and type 3 cytokines, JAK inhibitors are potentially efficacious for Th1/Th2/Th17-dominant and Th2/eosinophil-inferior endotypes. In this sense, it is anticipated that JAL inhibitors exert therapeutic effects on the broad spectrum of AD.

## 7. Conclusions

The subtypes of AD have been proposed through different conceptual ways. The classical subtypes are dubious, but clinically useful in daily outpatient care. The phenotype, representing the skin manifestation, is an ordinarily utilized technique for the diagnosis of AD. However, it cannot efficaciously contribute to subtype division, because the vast majority of skin manifestations is commonly seen among AD patients. The endotype is beneficial for more accurately subtyping AD patients, although its clinical use is a future issue. The relationship between the classical subtype, phenotype, and endotype is not simple, but rather complicated. For example, extrinsic AD, a representative classical subtype of AD, shows dry skin, palmar hyperlinearity, and ichthyosis vulgaris, which are the phenotypes. Given that these skin lesions are caused by *FLG* mutations, extrinsic AD manifests the endotype. Thus, the endotype can be implicated in some of the classical subtypes. We therefore tried to resolve the issue of how the classical subtypes correspond to the endotypes in AD, and Tokura Y presented the essence of this study in the International Eczema Council at the International Society for Investigative Dermatology Meeting held in 2023.

Recent findings have revealed the importance of the endotype for classification, as we can more accurately characterize each patient by using the endotype. To choose the therapeutic option, such as cytokine- or receptor-targeted biologics and JAK inhibitors, information on AD endotypes may be helpful. A good example is the cytokine-based endotype, which is categorized based on type 2, type 1, and/or type 3 cytokine-high properties [1,2,3,4]. In a comparison with the type 3-dominant condition of psoriasis, the range of immune dysregulation in AD is uniquely wide, suggesting that a personalized therapeutic approach is critical in AD patients. Many cell populations are involved in AD pathogenesis, including acquired and innate immune cells. Acquired immune cells, i.e., Th2, Th1, Th17, and Th22 have been studied in both peripheral blood and skin, and the findings were used for AD subtyping, such as extrinsic and intrinsic AD. Recently, innate immune cells have been more interesting as ILC2 frequency and activity can be a useful viewpoint for AD endotyping. Such a study would be conducted in future.

On one hand, the current endotype approach of AD has further clarified that AD is a heterogeneous and complicated disorder. On the other hand, investigators should reconsider the validity of criteria of AD, because endotype analysis is performed in certain groups that are defined as AD with one of the classical criteria [13,56,57]. Given that the AD criteria are based on the phenotype, disease history, and comorbidities, the current endotype studies have been conducted in limited patients classically diagnosed as having AD with their phenotype and allergic history. This raises the possibility that some endotypically important AD groups have been excluded by the potentially biased criteria. In this context, it is interesting to see whether endotypes or clusters would be present in other eczematous dermatitis conditions such as contact dermatitis [58]. A combination of endotypic signs with the classical subtypes may currently be a realistic way to choose individualized treatment options, but the further development of endotypic analyses may clarify the therapeutic approach to AD.

## 8. Limitations

Although there have been a considerable number of articles reporting AD endotypes, studies on the relationship between the classical subtype and the endotype have been scarcely reported. Therefore, we believe that our article provides a new aspect of AD studies.

However, we have some difficulties in searching reported articles with the key words, i.e., classical subtypes and endotypes. As a limitation of our article, therefore, it is not strongly convincing that the information presented here reflects current knowledge and research and that important aspects of endotypes are revealed.

Cytokine-based endotyping suggests that type 2 cytokine-targeting treatments are theoretically more effective for extrinsic, European–American and pediatric AD than intrinsic, Asian, and adult AD, respectively. The reality is, however, that they are also effective for intrinsic, Asian, and adult AD in most cases [59,60]. Their therapeutic efficacies should be compared between the classical subtype counterparts in future studies.

## Figures and Tables

**Figure 1 ijms-25-00265-f001:**
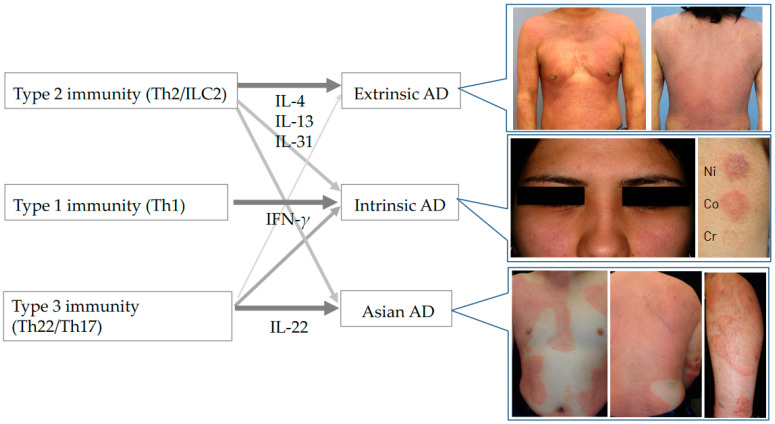
Cytokine patterns in extrinsic/intrinsic AD and Asian AD. Clinical photographs showing representative patients. Extrinsic AD: a 31-year-old man with a serum IgE level of 15,231 kU/L. Intrinsic AD: a 22-year-old woman with a serum IgE level of 11 kU/L, showing a nickel- and cobalt-positive patch test. Asian type of AD: a 31-year-old Japanese man with a serum IgE level of 22,216 kU/L, showing well-demarcated eczematous lesions.

**Figure 2 ijms-25-00265-f002:**
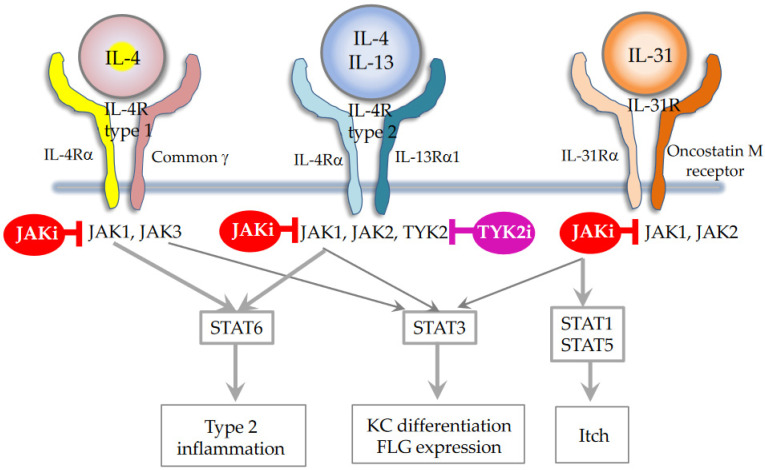
Type 2 cytokines and JAK/STAT signaling. Modified from References [7,54].

**Figure 3 ijms-25-00265-f003:**
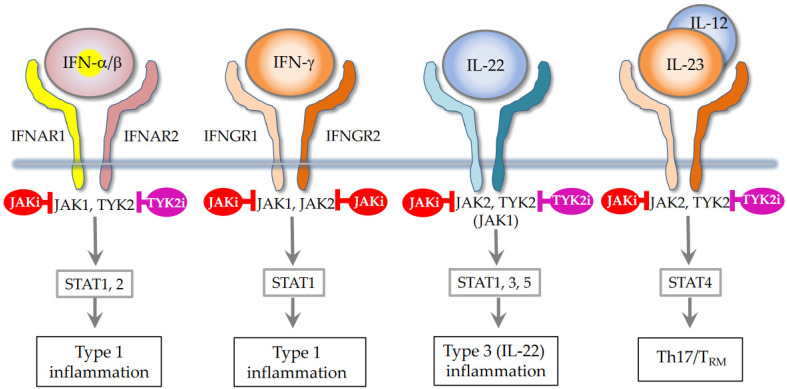
Type 1/Type 3 cytokines and JAK/STAT signaling. Modified Reference [50].

**Table 1 ijms-25-00265-t001:** Classical AD subtypes.

Subtype	Subtyping Approach	T Cell/Cytokine Skewing
Extrinsic and intrinsic subtypes	Serum IgE Levels	Th2 vs. Additional Th1/T22/Th17
European–American and Asian subtypes	Ethnicity	Th2 vs. Additional Th17/IL-19
Pediatric and adult subtypes	Age bracket	Th2 vs. Th1/CLA^−^ Th2/Th17/Th22

**Table 2 ijms-25-00265-t002:** Viewpoints and methodologies for endotyping AD.

Classification Viewpoints	Endotypes
Cytokines of acquired immunity	Type 2 cytokine-high Type 1 cytokine-high IL-22 (IL-17)-high or mixed
Allergens	Protein vs. metal/hapten
Epidermal barrier	Impaired vs. preserved
Intercellular lipids	Ceramide variation
Innate immunity	Strong vs. weak involvement
Serum biomarkers 143 markers (Ref. [34])	Cluster A: Skin-homing chemokines/IL-1R1-dominant Cluster B: Th1/Th2/Th17-dominant Cluster C: Th2/Th22/PARC-dominant Cluster D: Th2/eosinophil-inferior

## Data Availability

No new data were created or analyzed in this study. Data sharing is not applicable to this article.

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
