# Peer review of "How Do Classical Subtypes Correspond to Endotypes in Atopic Dermatitis?"

_ijms, 2023, doi:10.3390/ijms25010265_

Round 1

Reviewer 1 Report

Comments and Suggestions for Authors

The authors drafted a review focusing on endotypes of atopic dermatitis (AD). The article proposed that endotypes of AD could be stratified by several different point of views, including cytokine expression patterns, allergen properties, epidermal barrier conditions, and so on. The authors thought that the cytokine-based endotype seemed be the most useful one and was categorized into type 2 cytokine (IL-4, IL-13 and IL-31)-high, type 1 cytokine (interferon-g)-high, and/or type 3 cytokine (IL-22 and IL-17)-high, or mixed subtypes. These endotypes of AD were prerequisite for personalized therapeutic choices. The article was well-organized and comprehensive. I have only one suggestion.

1. Could the authors discuss more about the limitations of these AD endotypes, especially the cytokine-based endotyping? For example, though a treatment based on the cytokine expression pattern is theoretically more effective, a Th2-targeting treatment works well in most of the cases, including patients of Asian AD.  

Reviewer 2 Report

Comments and Suggestions for Authors

The authors provide a narrative overview on a "hot topic" in atopic dermatitis, that is the endotypic classification of the disease.

Overall the work is well-written but in my opinion it seems that nothing really new is added with this overview. The title should be changed to better focus on the narrative viewpoint of the present work: I don't see a clear correspondence between the endotypes and phenotypes.

The key point is missing in the conclusion, which appears a bit confusionary.

The AD phenotypes described in chapter 3 are mainly based on the Hannifin and Rajka classification, but in recent years others classification have been proposed (e.g. prurigo nodularis as a subtype of AD, early-onset and late onset -which are slightly different to the pediatric and adult types sub-classification-). AD classification is intricate, but for completion also other classification systems should be mentioned. 

In chapter 5 the different viewpoints of endotypes are presented, but a clear relationship with the clinical phenotypes is not apparent. Moreover some cytokines are actually serum biomarkers, and they are treated separately in two sub-chapters.

The authors should make an effort to homogenize the whole manuscript in order to clearly establish a correspondence, basing on the current data, between phenotypes and endotypes providing also a personal viewpoint. On the contrary the manuscript will appear as a simple list.

Comments on the Quality of English Language

Minor english editing and grammar/sentence check should be performed.

Reviewer 3 Report

Comments and Suggestions for Authors

Comments:

1)      Section 2, Methodology should clearly defined the basic terms i.e. , phenotypes, genotypes, regiotypes and theratypes, how these terms differ and may be applied to AD?

2)      Sections are not clearly described and there is lack clear demarcation of endotypes suggested by authors, its overlapping e.g. innate immunity vs cytokines.

3)      Some endotypes e.g. innate immunity? on what basis innate immunity can be defined as weak vs strong, microbiome? Table 2 mentioned microbiome? Not defined in the text, cite appropriate reference to support microbiome based endo types.  

4)      Authors tried to define Endotypes and suggested biomarkers. But authors not defined endotypes are related to tissue expression or in peripheral circulation (e.g. serum).

5)      There are various immune cell involvements in AD, it will be interesting if authors can suggest endotypes based on the frequency of immune cell populations.

6)      The study will be more informative and convincing if authors performed meta-analysis of available data to support the reliability of the endotypes in AD.

Comments on the Quality of English Language

May require attention in Typos and reconsideration/reframing some sentences.

Reviewer 4 Report

Comments and Suggestions for Authors

The review is on a topical subject and may be useful to readers of the magazine. However, I have a few comments:

1. Introduction. I suggest that the authors should be clearer about the aims and objectives of the review. It is necessary to specify the databases used, as well as the selection methodology, namely the exclusion/inclusion of articles. It seems that the submitted manuscript is a "narrative review". In particular, it is unclear how the 54 scientific articles presented in the manuscript were selected, whether a process of systematic review of the literature was carried out.

2. How the authors ensured that the information presented in the manuscript reflected current knowledge and research and that important aspects were not overlooked. This should be described or discussed in the limitations section. It is unclear whether the authors have found previous reports/studies that do not fit or support the proposed model. Have such studies been considered in this paper?

3. Conclusions. Conclusions and implications for future research are limited in this section of the manuscript.

4. General recommendations regarding the methodology of the review. Although the topic and work are of potential interest to the field, the authors may wish to consider a more systematic approach to a particular topic.

Other minor comments:

5. Abstract. (Line 21) It says "interferon-g", it should be "IFN-γ".

6. (Line 96). It is advisable to indicate that IL-19 can be related to type 3 immune response (mainly in the presence of autoimmune mechanisms), but also to type 2 (in the Th2-dependent allergic process).

Round 2

Reviewer 2 Report

Comments and Suggestions for Authors

Dear authors, I appreciate the efforts put in the revised version of the manuscript. 

The topic is difficult to approach given the great heterogeneity of the studies and the different classifications and viewpoints used for describing AD, however the purposes of the paper have been clarified and the differences between "subtypes", "endotypes" and "phenotypes" are now clearly expressed and discussed throughout the manuscript. Issue resolved. I think that also the potential readers could benefit from the changes.

Regarding the title of the work and the overall structure, I had the occasion to read the other Reviewer comment during the review process and we both had the same impression on the manuscript but gave two opposite solutions:  to define systemicatically the data presented or to clearly express that the presented data are based on a personal, expert, viewpoint (my opinion).

As in the limitation section the Authors stated that the study was based on a talk in the International Eczema Council I think that this article should be considered a viewpoint (and this should also be highlighted in the title of the article, the introduction and the conclusion).

A systematic review on the topic has its own difficulties, but in the present form and declared purposes this is not a systematic review: however the searching terms are correct and give the whole article more solidity.

Comments on the Quality of English Language

Moderate english check to be performed.

Author Response

Dear authors, I appreciate the efforts put in the revised version of the manuscript. 

The topic is difficult to approach given the great heterogeneity of the studies and the different classifications and viewpoints used for describing AD, however the purposes of the paper have been clarified and the differences between "subtypes", "endotypes" and "phenotypes" are now clearly expressed and discussed throughout the manuscript. Issue resolved. I think that also the potential readers could benefit from the changes.

Response:

We appreciate the Reviewer for the understanding.

Regarding the title of the work and the overall structure, I had the occasion to read the other Reviewer comment during the review process and we both had the same impression on the manuscript but gave two opposite solutions:  to define systemicatically the data presented or to clearly express that the presented data are based on a personal, expert, viewpoint (my opinion).

Response:

We agree with the Reviewer. A better solution is to clearly express that the presented data are based on a personal, expert, viewpoint. We have added the following sentences in the text (Introduction, last paragraph).

“Because of the great heterogeneity of the studies and the different classifications and viewpoints used for describing AD, it is difficult to define systemicatically the reported data. Therefore, our better solution is to clearly express that the presented data are based on a personal, expert, viewpoint.”

As in the limitation section the Authors stated that the study was based on a talk in the International Eczema Council. I think that this article should be considered a viewpoint (and this should also be highlighted in the title of the article, the introduction and the conclusion).

Response:

We have moved the following notion to the Introduction, 2nd paragraph,

“In this line of study, Tokura Y presented our viewpoint in International Eczema Council at the International Society for Investigative Dermatology Meeting held on May 10, 2023 in Tokyo, with open discussion.”

In addition, we have added the following line to the Conclusions, 1st paragraph.

“…and Tokura Y presented the essence of this study in International Eczema Council at the International Society for Investigative Dermatology Meeting held in 2023.”

A systematic review on the topic has its own difficulties, but in the present form and declared purposes this is not a systematic review: however the searching terms are correct and give the whole article more solidity.

Response:

We thank the Reviewer for properly understanding.

Reviewer 4 Report

Comments and Suggestions for Authors

The authors have responded appropriately to the comments. In my opinion, the article can be published in this form.

Author Response

We thank the Reviewer. 

Round 3

Reviewer 2 Report

Comments and Suggestions for Authors

The authors addressed my comments and suggestions. No other comments from my side.

Comments on the Quality of English Language

Final check to be performed.